# A Novel Adaptive Flexible Capacitive Sensor for Accurate Intravenous Fluid Monitoring in Clinical Settings

**DOI:** 10.3390/s25144524

**Published:** 2025-07-21

**Authors:** Yang He, Fangfang Yang, Pengxuan Wei, Zongmin Lv, Yinghong Zhang

**Affiliations:** School of Electromechanical Engineering, Guilin University of Electronic Technology, Guilin 541004, China; 19840432764@163.com (Y.H.); 15778303590@163.com (F.Y.); 17331972822@163.com (P.W.); lvzongmin0313@163.com (Z.L.)

**Keywords:** double pole plate, flexible capacitive sensing, adaptive, calibration-free

## Abstract

Intravenous infusion is an important clinical medical intervention, and its safety is critical to patient recovery. To mitigate the elevated risk of complications (e.g., air embolism) arising from delayed response to infusion endpoints, this paper designs a flexible double pole capacitive (FPB) sensor, which includes a main pole plate, an adaptive pole plate, and a back shielding electrode. The sensor establishes a mapping between residual liquid volume in the infusion bottle and its equivalent capacitance, enabling a non-contact adaptive monitoring system. The system enables precise quantification of residual liquid levels, suppressing baseline drift induced by environmental temperature/humidity fluctuations and container variations via an adaptive algorithm, without requiring manual calibration, and overcomes the limitations of traditional rigid sensors when adapting to curved containers. Experimental results showed that the system achieved an overall sensitivity of 753.5 fF/mm, main pole plate linearity of 1.99%, and adaptive pole plate linearity of 0.53% across different test subjects, linearity of 0.53% across different test subjects, with liquid level resolution accuracy reaching 1 mm. These results validate the system’s ultra-high resolution (1 mm) and robust adaptability.

## 1. Introduction

Intravenous (IV) infusion is regarded as a clinical “lifeline”, with a high utilization rate of 71.9–95.0% in hospitalized patients [1], and is an extremely important therapeutic tool. However, IV infusion harbors many medical care safety hazards, and problems such as air embolism, phlebitis, and oozing are common. Among them, air embolism has a high lethality rate, and failure to treat promptly when the infusion is incomplete is one of the main reasons for this serious condition. Currently, liquid level monitoring technology is one of the effective solutions to prevent air embolism caused by empty infusion bottles. However, the existing technologies still have problems such as unclear working principles and limited use conditions, which make it difficult to achieve accurate early warning before the end of infusion.

Currently, liquid level monitoring techniques can be categorized into two systems: contact and non-contact [2]. Among the contact methods are the following: (1) fiber optic level monitoring [3,4,5]: based on the optical signal modulation principle, with anti-electromagnetic interference capability and high temperature and pressure resistance, suitable for monitoring chemically corrosive liquids, but its measurement accuracy is susceptible to liquid adhesion interference; (2) electrode level monitoring [6]: achieving level triggering through the conductive liquid and electrode, with advantages of low cost and fast response, but only applicable to conductive media and with the risk of electrode contamination. Non-contact technologies include the following: (1) microwave level monitoring [7]: using the microwave reflection principle to achieve ±1 mm high-precision-level measurement, able to adapt to extreme environments, but its signal attenuation is sensitive to the low dielectric constant of the liquid and the equipment cost is higher; (2) optical level monitoring [8,9]: based on the characteristics of light reflection/refraction, applicable to rapid monitoring of transparent liquids, but bubbles or vapors will lead to measurement deviation; (3) image level monitoring [10,11]: analyzing the liquid surface morphology through machine vision, capable of handling complex dynamic scenes, but its performance depends on lighting conditions and has a computational load issue; (4) capacitance level monitoring [12,13,14]: based on the principle that the liquid level change causes a change in the effective area of the electrodes, with a compact structure, but temperature, humidity, and container shape will affect the measurement accuracy; (5) ultrasonic level monitoring [15,16]: based on the principle of acoustic time-of-flight, with advantages of convenient installation and high cost-effectiveness, but requiring temperature adaptation and being susceptible to foam interference. All of the above methods have their own advantages, but in specific application scenarios (e.g., intravenous fluid (IV) level monitoring), these methods still face limitations such as high technical complexity, expensive cost, or dependence on specific environmental conditions (e.g., light).

Current IV level monitoring technologies face multiple technical bottlenecks: the CD4-LLS capacitive device developed by Zhongyue Tang’s team [17] has a wide range (900 mm) and high linearity, but it relies on a manual pre-calibration process; N. Giaquinto et al.’s [18] deep-learning-based multiview vision system achieves full-scene recognition but has a cost-robustness imbalance due to high computing power requirements and sensitivity to the light environment; Lee J-K et al. [19] used a mechanical–optical multi-sensor fusion strategy to increase the dynamic droplet monitoring rate to 88%, but the multi-sensor architecture significantly increased the system power consumption and integration complexity; Wei-Hsiung Tseng et al. [20] designed a biconvex lens light-focusing module to optimize sensitivity, but the precision optical components pose miniaturization design challenges and the risk of lens contamination. None of the above studies have effectively addressed the synergistic optimization among environmental interference suppression, cost constraints, and measurement reliability in medical scenarios. A comparative analysis between the aforementioned IV monitoring schemes and the IV monitoring system based on a flexible (FPC) double pole capacitive sensor proposed in this study is presented in Table 1.

To address the measurement limitations of the existing intravenous infusion monitoring technologies, this paper designs a flexible (FPC) coplanar double plates capacitance sensor based on the parallel-plate capacitance sensing mechanism, and establishes the mapping relationship between the residual amount of liquid in the infusion bottle and the capacitance analog by deploying two metal detection pole plates of different sizes on the front side of the flexible material and metal integrating shielding electrodes on the back side. On this basis, an adaptive intravenous infusion monitoring system was constructed. The system’s high-precision capacitance detection circuit paired with the adaptive algorithm realizes the conversion of capacitance analog quantity to the height data of liquid residual quantity, effectively eliminates the influence brought by disturbing factors such as the physical properties of infusion bottles and differences in the dielectric properties of solutions, and significantly improves the universality of level detection for multi-scenarios and multi-dosage forms of intravenous fluids. Finally, five types of clinically used intravenous fluids are selected for multi-group parallel control experiments. The experimental results show that the system has high-precision measurement and self-adaptive capabilities and can meet the demand for non-contact detection of intravenous fluids in clinical medicine.

## 2. Theoretical Principles

Based on classical electromagnetic field theory, a parallel-plate capacitor is fundamentally configured with two parallel conductive pole plates subjected to a potential difference, between which a quasi-static electric-field energy-storing system is established via dielectric isolation [21]. As the core parameter characterizing charge storage capability, capacitance is positively proportional to the effective overlapping area of the pole plates. Leveraging this physical property, a parallel-plate capacitive structure is formed through electric-field coupling among three components: the intravenous fluid (acting as a movable pole plate, since clinical intravenous infusions primarily artificially formulated electrolyte solutions contain abundant free ions from dissolution, rendering them highly conductive), the composite dielectric layer comprising the insulated infusion container wall and an adhesive layer (the latter ensures intimate contact between the sensor and container wall, eliminating air gaps and suppressing stray capacitance to enhance measurement accuracy), and the capacitance sensor (serving as the fixed pole plate). Consequently, the capacitance magnitude is proportional to the height of the residual intravenous fluid covering the pole plate. Furthermore, the back surface of the capacitance sensor integrates shielding electrodes to mitigate the influence of parasitic capacitance and ambient interfering electric fields [22].

The detection principle of the capacitance sensor is schematically illustrated in Figure 1. The main pole plate forms a capacitance $C_{1}\;$through coupling with the intravenous fluid, while the adaptive pole plate forms $C_{2}$ via coupling with the same fluid; the shielding electrode forms capacitances $C_{3}\;$and${\;C}_{4}$ through coupling with the main pole plate and the adaptive pole plate, respectively. Notably, $C_{3}\;$and${\;C}_{4}$ assist in suppressing stray capacitance but are negligible in theoretical analyses [23]. Importantly, extraneous interference from coplanar capacitive coupling is avoided by employing a common excitation source for both the main pole plate and adaptive pole plate [24].

Based on the physical model of the capacitance sensor outlined above, we proceed to derive the mathematical model for capacitance measurement. Let the width of the pole plate be w, the height of the remaining IV fluid covering the pole plate be h, and the effective area be A. Since the flexible pole plate is tightly adhered to the container wall, the effective areas of capacitances $C_{1}$ and${\;C}_{2\;}$ correspond exactly to the area wetted by the IV fluid, i.e.,(1)A=wh

From the parallel-plate capacitance equation,(2)$$C=\frac{\varepsilon_{0}\varepsilon_{e}A}{d}$$
where${\;\varepsilon }_{0}\;$is the vacuum permittivity,$\;\varepsilon_{e}\;$is the relative permittivity of the inter-plate dielectric, and d the plate separation. For the composite dielectric layer (adhesive + IV container wall), the equivalent relative permittivity is as follows:(3)$$\varepsilon_{e}=\frac{d_{s}+d_{j}}{\frac{d_{s}}{\varepsilon_{s}}+\frac{d_{j}}{\varepsilon_{j}}}$$

Here,${\;\varepsilon }_{s\;}$,${\;d}_{s}\;$are the adhesive layer’s permittivity and thickness;$\;\varepsilon_{j\;}$,$\;d_{j}\;$are the container wall’s permittivity and thickness. Under ideal conditions, capacitance is thus:(4)$$C_{t}=\frac{\varepsilon_{0}\varepsilon_{e}wh}{d_{s}+d_{j}\;}$$

The above derivation assumes ideal conditions with a uniform electric-field distribution across the pole plate. In reality, edge electric fields are non-uniform (Figure 2).

To mitigate fringing effects, a parallel compensation capacitance${\;C}_{e\;}$is introduced [25]. Let${\;C}_{s}\;$denote the actual measured capacitance; the corrected expression from Equation (4) is as follows:(5)$$\begin{array}{ll} C_{s} & =C_{t}+C_{e}\\ & =\frac{\varepsilon_{0}\varepsilon_{e}wh}{d_{s}+d_{j}}+\frac{\varepsilon_{e}h}{\pi }\left\{1+\ln\left[1+\frac{2\pi w}{d_{s}+d_{j}}+\ln\left(1+\frac{2\pi w}{d_{s}+d_{j}}\right)\right]\right\}+\\ & \frac{\varepsilon_{e}w}{\pi }\left\{1+\ln\left[1+\frac{2\pi h}{d_{s}+d_{j}}+\ln\left(1+\frac{2\pi h}{d_{s}+d_{j}}\right)\right]\right\} \end{array}$$

Equation (5) describes the sensor–fluid coupling capacitance, extending the parallel-plate model with corrections for composite dielectric permittivity and edge field distribution. It quantifies capacitance–physical parameter relationships, integrating container wall dielectric properties and edge effects to characterize the mapping between remaining fluid height and output capacitance.

## 3. Sensor Design and Fabrication

This section elaborates on the sensor design. The sensor utilizes polyimide (PI) as the flexible substrate and 3M double-sided tape as the adhesive. Polyimide exhibits excellent insulation, flexibility, chemical stability, low thermal expansion coefficient, and low hygroscopicity—properties that significantly mitigate environmental interference [26].

Given the variability in environmental parameters across intravenous fluid containers, fixing the environmental parameters of a single pole plate (per Equation (5)) would introduce measurement errors for different objects due to parameter fluctuations. To address this, two metal pole plates (identical width, varying lengths) are positioned on the same horizontal plane of the flexible substrate, arranged vertically in parallel with a 2 mm gap (a spacing that does not impact performance). One serves as an adaptive pole plate to calibrate environmental parameters, enabling dynamic adjustment to environmental changes; the other functions as the main pole plate to monitor the remaining intravenous fluid level based on the environmental parameters measured by the adaptive pole plate. The initial position and total length of the main pole plate outside the bottle determine the starting value and total range of fluid level monitoring.

To ensure the sensor accurately measures the remaining amount of intravenous drug solution, the sensor’s tip must be closely positioned near the upper part of the intravenous infusion bottle’s mouth. Therefore, the sensor’s width shall not exceed the outer diameter of the bottle mouth. Simultaneously, to maximize sensor sensitivity, its width should match the bottle’s outer diameter. By reviewing the data, the outer diameter of common IV bottles is found to be 20 mm. Considering the above constraints and the requirement for maximum sensitivity, the sensor width is set to 20 mm. According to the “IV Therapy Nursing Technical Code of Practice”, nurses should monitor the infusion situation within 6 min before the infusion ends. Considering the general drop rate (60 drops/min) and drop coefficient (15 drops/mL), the sensor is placed at the top of the remaining liquid. Calculations show that the sensor is triggered when the remaining liquid is 24 mL, so the length of the main pole plate is set to 24 mm.

The adaptive pole plate is used for environmental parameter measurement, with its length positively correlated to measurement accuracy. Comparative experiments (main pole plate: 24 mm × 20 mm; adaptive pole plate: 18 mm width with lengths of 1, 2, 3, and 4 mm) confirmed that a 4 mm length results in a relative error between the integrated environmental coefficients of the adaptive and main pole plates of less than 1%, meeting system requirements. Experimental results are presented in Section IV.

In summary, the main pole plate is designed as 24 mm × 20 mm, and the adaptive pole plate as 4 mm × 20 mm. A 20 mm × 30 mm rectangular shielding electrode is integrated on the back of the flexible substrate to isolate electromagnetic interference. Electrical connection between the sensor and main control board is achieved via a connector with an array of metal pins (4 mm length, 1 mm width, 1 mm pitch). The back of the pole plates is reinforced with PI to enhance mechanical strength, while 3M double-sided tape ensures firm adhesion to the bottle. Sensor prototypes, fabricated by JLCPCB (Shenzhen, China) and shown in Figure 3, utilize a mature FPC manufacturing process suitable for mass production with competitive costs.

## 4. Detection System Design

### 4.1. Hardware Design

The FDC2214, a four-channel high-precision capacitance digitizer from Texas Instruments (TI, Dallas, TX, USA), is designed for non-contact sensing in complex environments. Based on LC resonance principles, it converts capacitance variations induced by the target into digital outputs. With 28-bit resolution, support for four-channel multi-node monitoring, and a maximum sampling rate of 13.3 ksps, its innovative anti-jamming design integrates differential sensing and electromagnetic shielding to suppress ambient noise, ensuring stable performance under temperature fluctuations, humidity changes, or pollutant exposure. The simplified circuit of the FDC2214 (from its datasheet) is shown in Figure 4.

In Figure 4,$\;C_{1}$ and$\;C_{2}\;$represent the equivalent capacitances formed by the main and adaptive pole plates with the intravenous fluid, respectively. According to the FDC2214 chip manual, their values are derived using the following:(6)$$C_{x}=\frac{1}{{L(2\pi f_{sensor})}^{2}}$$
where$\;f_{sensor}\;$is the resonant frequency, L is the 18 μH inductance in the LC oscillation circuit, and $C_{x}\;$(including the 33 pF capacitor *C*) is the measured capacitance. According to the FDC2214 chip manual, the resonant frequency is calculated as follows:(7)$$f_{sensor}=\frac{f_{REF}\cdot DATA}{2^{28}}$$
with${\;f}_{REF}\;$being the 40 MHz crystal oscillator and DATA the system-acquired digital capacitance value. Substituting Equation (7) into Equation (6) yields the following:(8)$$C_{x}=\frac{1}{L}{\left(\frac{2^{28}}{2\pi f_{REF}\cdot DATA}\right)}^{2}$$

Thus, the analog capacitance from the sensor is digitized by the FDC2214, and the coupling capacitance with the intravenous fluid is derived via Equation (8). The hardware design block diagram and main control board are shown in Figure 5 and Figure 6, respectively.

The ESP32-WROOM microprocessor serves as the core control unit, real-time collecting digital signals from flexible capacitance sensors via the $I^{2}C$ bus. These signals are processed by adaptive algorithms to predict the remaining fluid height and infusion completion time. Structured data are transmitted to mobile monitoring platforms (e.g., mobile apps) using the MQTT protocol. When the liquid level drops below the safety threshold, a graded alarm is activated: priority alarms are first sent to the host computer; if unacknowledged within a preset delay, a GPIO-driven acoustic–optic module triggers on-site warnings via LED strobing and beeping, forming a closed-loop safety system.

### 4.2. Software Design

#### 4.2.1. Adaptive Algorithm Foundation Building

To ensure experimental rationality and accuracy, Equation (5) is optimized using the sensor’s actual dimensions (Section 3). Given the flexible capacitive sensor’s main pole plate length = 24 mm, adaptive pole plate length = 4 mm, and maximum height of remaining IV fluid covering the pole plate = 24 mm, we analyze Equation (5) for$\;h\in \left(0,\;24\right)$. We define the following:(9)$$f\left(h\right)=\ln\left[1+\frac{2\pi h}{d_{s}+d_{j}}+\ln\left(1+\frac{2\pi h}{d_{s}+d_{j}}\right)\right]$$

For h→0, $f\left(h\right)\;$is Taylor-expanded to first order. Within the range of h, the ratio of${\;C}_{s}\;$before and after expansion is approximately 1, allowing Equation (5) to be simplified as follows:(10)$$C_{s}\approx \left\{\frac{\varepsilon_{0}\varepsilon_{e}w}{d_{s}+d_{j}}+\frac{\varepsilon_{e}}{\pi }\left\{1+\ln\left[1+\frac{2\pi w}{d_{s}+d_{j}}+\ln\left(1+\frac{2\pi w}{d_{s}+d_{j}}\right)\right]\right\}+\frac{4\varepsilon_{e}w}{d_{s}+d_{j}}\right\}h+\frac{\varepsilon_{e}w}{\pi }$$

We then take into account the influence of parasitic capacitance${\;C}_{j}\;$(from components like the circuit board and shielding layer), which is in parallel with${\;C}_{s}$, requires Equation (10) to be corrected as follows:(11)$$C_{se}\approx \left\{\frac{\varepsilon_{0}\varepsilon_{e}w}{d_{s}+d_{j}}+\frac{\varepsilon_{e}}{\pi }\left\{1+\ln\left[1+\frac{2\pi w}{d_{s}+d_{j}}+\ln\left(1+\frac{2\pi w}{d_{s}+d_{j}}\right)\right]\right\}+\frac{4\varepsilon_{e}w}{d_{s}+d_{j}}\right\}h+\frac{\varepsilon_{e}w}{\pi }+C_{j}$$

Sensor sensitivity, derived from Equation (11), is as follows:(12)$$k=\frac{∆C_{se}}{∆h}$$

The above is the mathematical model expression of Equation (5) after simplification based on the physical model of the flexible capacitive sensor. Through the derivation of the sensor’s mathematical model, it can be concluded that the sensor exhibits the following three characteristics: First, from Equation (2), as the distance between the sensor and the intravenous fluid is much smaller than the effective area of the pole plate, the initial capacitance value will vary due to differences in the wall thickness of different infusion containers and the dielectric constant of their materials. Second, from Equation (11), since the dielectric constant and geometrical dimensions of the flexible sensor are constant, the capacitances${\;C}_{1}\;$and${\;C}_{2}$ will change linearly with the height h of the fluid covering the pole plate. Finally, from Equation (12), the detection sensitivity of the sensor is influenced by the dielectric constant and thickness of the fluid container.

#### 4.2.2. Adaptive Algorithm Implementation

To meet the demand for real-time monitoring of clinical intravenous infusion, the liquid level height resolution of the remaining intravenous fluid covering the pole plate is calibrated to 1 mm (The FDC2214 capacitive digitizer equations 28-bit high resolution, enabling detection sensitivity at the sub-femtofarad (fF) level. When applied to other high-precision measurement scenarios, the system’s functionality can be migrated by reconstructing the signal processing algorithm at the software layer without requiring hardware architecture adjustments. Through Equation (11), the following was found: let$\;k=\frac{\varepsilon_{0}\varepsilon_{e}w}{d_{s}+d_{j}}+\frac{\varepsilon_{e}}{\pi }\left\{1+\ln\left[1+\frac{2\pi w}{d_{s}+d_{j}}+\ln\left(1+\frac{2\pi w}{d_{s}+d_{j}}\right)\right]\right\}+\frac{4\varepsilon_{e}w}{d_{s}+d_{j}}$, $d=\frac{\varepsilon_{e}w}{\pi }+C_{j}$, then Equation (8) can be expressed as a primary function with h as the independent variable:(13)$$C_{se}=kh+d$$

When the height of the intravenous fluid covering the sensor exceeds 26 mm, let$\;C_{2b\;}$and$\;C_{2e\;}$denote consecutive capacitance measurements acquired by the main control circuit via Equation (8) for the adaptive pole plate, with corresponding liquid heights${\;h}_{2b}\;$and$\;h_{2e}$(${\;h}_{2b}\;$>$\;h_{2e}$) covering the adaptive pole plate. From Equation (13),(14)$$C_{2b\;}-C_{2e\;}=k_{2}({\;h}_{2b}-{\;h}_{2s})$$

Let $∆C_{2}=C_{2b}-C_{2e}$, $∆h_{2}=h_{2b}-h_{2s}$, then Equation (14) can be rewritten as follows:(15)$$∆C_{2}=k_{2}∆h_{2}$$

According to Equation (15), once the measurement environment of the adaptive electrode is established, the physical properties of the measured object (e.g., the material, wall thickness, and dielectric constant of intravenous infusion bottles) and non-abrupt environmental factors (e.g., temperature, humidity, etc.) will be determined. During capacitance sampling at each sampling cycle, the error capacitance introduced by variations in the infusion bottle’s physical properties and non-abrupt environmental factors is captured as fixed parasitic capacitance in every sampling. Thus, the capacitance difference between two consecutive samples comprises only the capacitance variation caused by liquid level changes, while the parasitic capacitance resulting from the inherent physical properties of the infusion bottle and non-abrupt environmental factors is eliminated by the difference calculation. This ensures the detection system is immune to errors induced by discrepancies in the physical properties of the measured object and non-abrupt environmental factors, thereby achieving adaptive calibration.

Similarly, for the main pole plate with fluid height in$\;\left(0,\;24\right)\;mm$, the capacitance variation with liquid height is expressed as(16)$$∆C_{1}=k_{1}∆h_{1}$$

Equation (8) reveals that the environmental adaptation coefficient k is governed by the synergistic interaction of the three parameters$\;\varepsilon_{e}$,$\;d_{s}+d_{j},\;$and w. Given that the adaptive pole plate and the main pole plate are integrated into the same monitoring system, and their environmental action parameters exhibit synchronous variation characteristics, the equivalence of their environmental adaptation coefficients can be derived as$\;(k_{1}=k_{1})$. Under this equivalence constraint, the analytical expression for the height ${\;h}_{w}\;$of the remaining liquid in the intravenous infusion is given by the following:(17)$$h_{w}=24-\frac{∆C_{1}}{k_{2}}$$

Furthermore, the system detection process is outlined as follows: when the height of the intravenous medicinal fluid covering the flexible capacitive sensor falls within the interval$\;h_{w}\in \left[26,\;30\right]\;$(unit: mm), the system activates an adaptive learning mode, sampling and storing the capacitance value of the adaptive pole plate at a 1 s interval. When the height reaches$\;h_{w}\in \left[24,\;26\right]$ (unit: mm), least-squares fitting is performed to calculate the environmental coefficient using historical data from the adaptive pole plate. When the height is$\;h_{w}\in \left[0,\;24\right]$ (unit: mm), this coefficient is combined with Equation (17) to achieve precise calculation of the liquid level height of the remaining intravenous drug solution, which is then transmitted to the mobile APP via the CPU.

The liquid level state transition triggers a multi-stage data processing mechanism: at the initial infusion stage, the sensor is fully submerged, leading to a capacitance steady state. Upon detecting the first-order capacitance change, the system identifies the height${\;h}_{w}\in \left[26,\;30\right]\;$(unit: mm). Leveraging the uncoated copper buffer strip between the pole plates, when the detected capacitance enters a secondary steady state, it indicates that the adaptive pole plate has detached from the liquid surface at height $h_{w}\in \left[24,\;26\right]$ (unit: mm). The subsequent dynamic capacitance response corresponds to the height${\;h}_{w}\in \left[0,\;24\right]$ (unit: mm). This multi-threshold decision strategy ensures the robustness of liquid level detection by integrating hardware topology characterization and signal pattern recognition.

Consequently, the model effectively mitigates the impact of environmental parameter fluctuations on the measurement system by continuously collecting capacitance differences detected by the flexible capacitive sensors for comparative analysis. This approach enables the measurement of the remaining fluid height and predicts the completion time of the infusion based on the rate of capacitance change. Moreover, due to the adoption of differential processing, the system remains unaffected by non-instantaneous environmental factors such as temperature, humidity, and parasitic capacitance.

## 5. Experiment

Having described the working mechanism of the IV infusion testing system in the preceding section, this section establishes an experimental environment consistent with typical indoor clinical settings (temperature: 18–26 °C; relative humidity: 40–60%) to align with real-world indoor clinical conditions, as shown in Figure 7a. Five representative intravenous solutions commonly used in clinical practice were selected as test substrates, depicted in Figure 7b, including 5% glucose injection, compound sodium chloride injection, calcium gluconate injection, 0.9% sodium chloride saline, and sodium lactate Ringer’s injection. These samples were chosen to cover variations in dielectric properties and ionic concentrations, facilitating a robust assessment of the system’s adaptability across clinically relevant fluid types. The experimental design is grounded in the correlation between the physical properties of different IV fluids/containers and sensor responses, constructing a generalized validation framework for infusion level detection.

All solutions were contained in polypropylene (PP) containers with a dielectric constant range of 2.2–2.3 and a wall thickness of 2–4 mm. To ensure experimental consistency, all tests were performed under controlled indoor conditions (temperature: 18–26 °C; relative humidity: 40–60%). Installation of the monitoring system involved two key steps: first, the sensor was firmly affixed to the outer container wall using 3M adhesive to ensure intimate contact; second, electrical interconnection between the sensor and main control board was established via a dedicated connector to enable signal transmission.

After system initialization, baseline capacitance measurements were performed for the five intravenous solutions under non-infusion conditions. During this phase, the fluid level completely covered both the main pole plate and adaptive pole plate, with 40 consecutive capacitance datasets collected at 1 s intervals. Given the discrete nature of initial capacitance values arising from differences in the measured media, a baseline normalization method was employed: the minimum capacitance value among the 40 datasets was designated as the reference, and the differences between the remaining measurements and this reference were used as analytical data. Notably, raw capacitance values included parasitic components from sources such as sensing electrodes and PCB traces. Table 2 summarizes the initial minimum capacitance values of the main and adaptive pole plates across different fluid conditions. After data processing, the capacitive response characteristics of each pole plate in various media are visualized in Figure 8, with separate plots for the main pole plate and adaptive pole plate.

Under standardized conditions, the sensor demonstrated distinct response patterns to different IV fluids. Specifically, the capacitance difference between the main and adaptive pole plates directly correlated with dielectric property variations arising from differences in container parameters. As shown in Figure 8, the deviation between the detected capacitance and reference capacitance for both pole plate types fluctuated within the range (0.1, 0.22) across all tested solutions, indicating a high degree of consistency. This outcome confirms that the sensor exhibits excellent reproducibility during routine operation in clinical environments. The observed differences primarily stem from intrinsic variations in container physical parameters, which align with the characteristics of the theoretical model described by Equation (2).

To validate the system’s monitoring performance during dynamic infusion, experiments were conducted by setting the flow rate to 60 drops/min (mimicking clinical routine) via a flow controller, with the device tracking liquid levels for the five distinct solutions. For each solution, continuous level measurements were recorded at 1 mL volume intervals by manually regulating fluid discharge to simulate real-time infusion dynamics; each solution was tested in five independent replicates to ensure statistical robustness. The measurement results are presented in Figure 9.

As illustrated in Figure 9, after five replicate tests for each solution, the system maintained stable detection accuracy under dynamic infusion conditions. Least-squares fitting of the capacitive responses yielded strong linear correlations (minimum $R^{2}>0.9935$), confirming the monitoring scheme’s superior robustness and linearity when handling diverse fluids in actual clinical infusion scenarios, which validates the correctness of Equation (13). Table 3 summarizes key parameters derived from the fitted curves: the slopes$\;k_{z}\;$(adaptive pole plate) and$\;k_{s}\;$(main pole plate), the relative error α between these slopes, and the linearity indices$\;\gamma_{z}\;$(adaptive pole plate) and$\;\gamma_{s}\;$(main pole plate). These parameters are defined as follows:(18)$$\alpha =\frac{\left|k_{s}-k_{z}\right|}{k_{s}}\times 100\%$$(19)$$\gamma =\frac{∆\beta }{\lambda }\times 100\%$$
where ∆β denotes the maximum absolute deviation between the fitted values and measured values, and λ represents the full-scale output range.

As shown in Table 3, the relative deviation of the linear slopes derived from least-squares fitting between the main and adaptive pole plate is less than 1%. This not only validates the correctness of Equation (3) in the theoretical model but also directly confirms the equivalence of their sensitivity coefficients. By excluding extreme values (maximum and minimum) from the slope dataset and calculating the arithmetic mean, this study determined the integrated sensor sensitivity as 753.5 fF/mm(corresponding to a 1 mm resolution) and derived the overall linearity of the main pole plate as $\gamma_{z}=1.99\%$ and that of the adaptive pole plate as $\gamma_{s}=0.35\%$. From a mechanistic perspective, the two electrodes exhibit differentiated graded linear characteristics within the effective measurement range: according to Equation (8), when the electrode width is fixed, the parasitic capacitance induced by the capacitive edge effect displays a nonlinear positive correlation with liquid coverage height. Given that the main pole plate length $d_{1}=24\;mm$ is substantially greater than the adaptive pole plate length $d_{2}=4\;mm$, the main pole plate accumulates a higher maximum parasitic capacitance, resulting in a significantly steeper slope in its fitted curve relative to the adaptive pole plate.

The aforementioned data comprehensively reveal the sensor’s dynamic monitoring performance: in intravenous infusion scenarios, as the main pole plate has a length of 24 mm, the maximum height measurement error introduced by the main pole plate’s overall linearity ($\gamma_{z}=1.99\%$) is less than 0.5 mm, which is far below the error range specified by clinical alarm thresholds—typically requiring alarms to trigger when residual liquid height is within 10 mm, with allowable error less than 1 mm. This enables near-instantaneous warning activation, ensuring infusion safety. Meanwhile, as the adaptive pole plate has a length of 4 mm, the measurement error (less than 0.05 mm) induced by its excellent overall linearity ($\gamma_{s}=0.35\%$) is negligible relative to the system’s 1 mm resolution, effectively mitigating interference from dynamic liquid surfaces on monitoring accuracy. Additionally, the maximum relative error in slope fitting between the two electrodes across five parallel experiments was only 0.66%, further validating the equivalence of their characteristics and strong consistency with the theoretical model. In summary, the sensor’s mathematical model has been experimentally validated, supporting the application of Equation (17) for precise measurement of residual intravenous drug height (dynamic monitoring results for the main pole plate coverage height are presented in Figure 10).

Figure 10 shows that during full-scale measurement, the difference between the height of the liquid covering the main pole plate as measured by the sensor and the actual liquid height is only 0.39 mm, fully meeting the accuracy requirements of the intravenous infusion detection system with a 1 mm resolution. The data processed by the algorithm and transmitted to the mobile phone APP via the CPU demonstrates that the displayed residual IV fluid height aligns perfectly with the actual value. In 25 parallel simulation experiments, the empty-bottle alarm successfully triggered with a 100% success rate.

Although this experiment only included five sets of parallel trials, the data indicate that the sensor’s measurements of residual fluid height exhibit high consistency across different intravenous solutions. The observed differences arise solely from the inherent physical properties of the IV fluid containers, which align perfectly with theoretical predictions. Thus, this system addresses critical limitations of existing technologies (e.g., eliminating manual calibration, adapting to curved containers) and meets clinical demands for high-precision IV monitoring. Its flexibility enables stable operation across diverse container sizes and geometries, supporting broad clinical applicability.

## 6. Conclusions

Intravenous infusion, a critical clinical “lifeline” utilized by 71.9–95.0% of hospitalized patients, poses substantial safety risks such as air embolism when infusion endpoints are managed belatedly. Whereas existing monitoring technologies—both contact-based (e.g., fiber optic, electrode) and non-contact (e.g., microwave, optical, imaging)—exhibit inherent limitations, including reliance on manual calibration, susceptibility to environmental disturbances (temperature, humidity, container variations), poor adaptability to curved containers, and high complexity or cost, this study has developed a non-contact adaptive flexible monitoring system for intravenous infusion based on flexible (FPC) capacitive sensors to address these unmet clinical demands. Characterized by a dual-pole plate configuration (a main pole plate for liquid level monitoring and an adaptive pole plate for environmental calibration), integrated with a back shielding electrode and an adaptive algorithm, the system eliminates errors arising from container physical properties and non-abrupt environmental factors without requiring manual calibration. Its flexibility further overcomes the rigidity constraint of traditional sensors. Experimental validation using five typical clinical intravenous solutions confirms its superior performance: a sensitivity of 753.5 fF/mm, a resolution of 1 mm, a maximum measurement error of 0.39 mm, a 100% success rate for empty-bottle alarms, and linearity of 1.99% (main pole plate) and 0.35% (adaptive pole plate), all of which fully meet clinical requirements. This research not only provides a reliable solution for accurate, real-time monitoring of intravenous fluids but also enhances infusion safety and clinical efficiency, thereby holding significant value for intelligent infusion management in clinical settings.

## Figures and Tables

**Figure 1 sensors-25-04524-f001:**
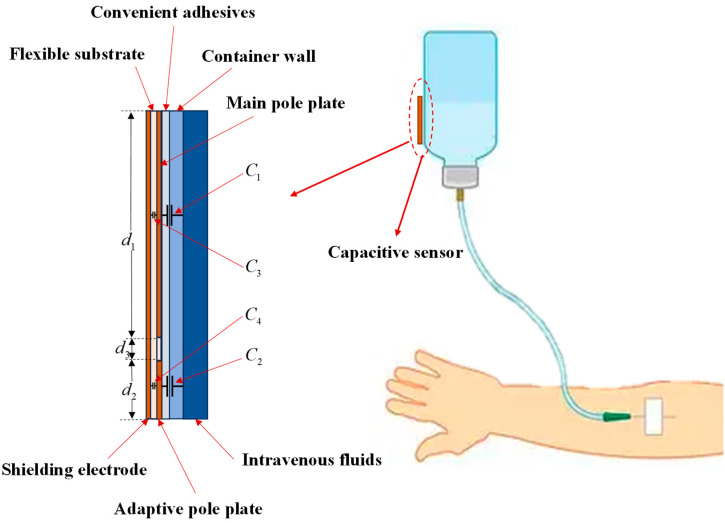
Schematic of the capacitive sensor detection principle.

**Figure 2 sensors-25-04524-f002:**
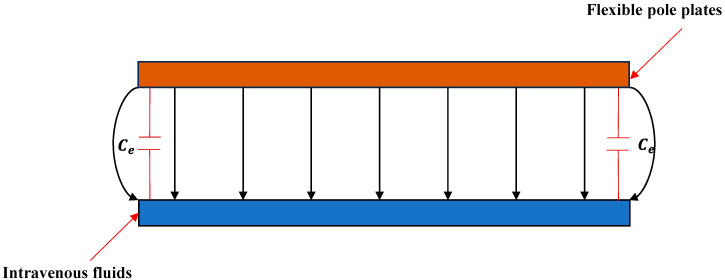
Schematic distribution of electric-field lines between the pole plates.

**Figure 3 sensors-25-04524-f003:**
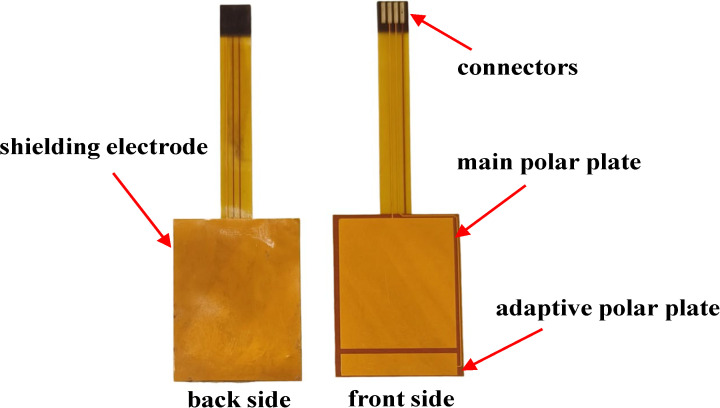
Actual image of flexible capacitive sensor.

**Figure 4 sensors-25-04524-f004:**
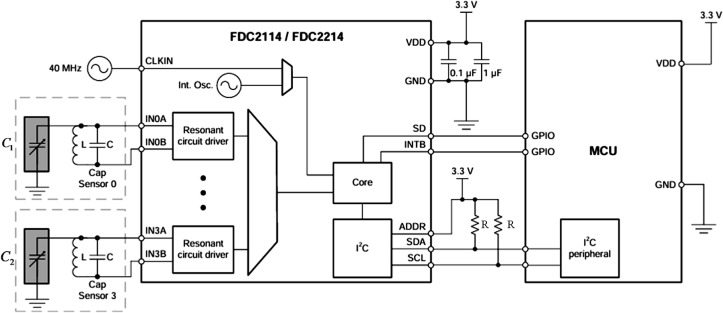
Simplified circuit diagram of FDC2214. Here, L denotes an inductor (18 μH), C a capacitor (33 pF), and R a resistor (10 kΩ). The black rectangular frame encloses the chip’s internal logic circuit schematic.

**Figure 5 sensors-25-04524-f005:**
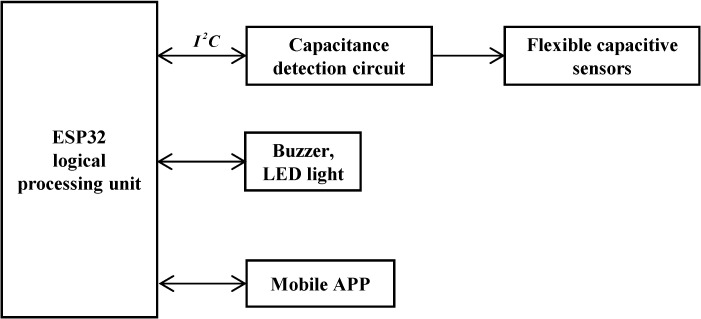
Block diagram of system structure.

**Figure 6 sensors-25-04524-f006:**
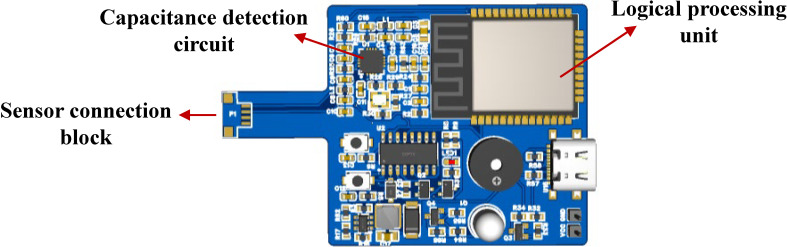
Main control board.

**Figure 7 sensors-25-04524-f007:**
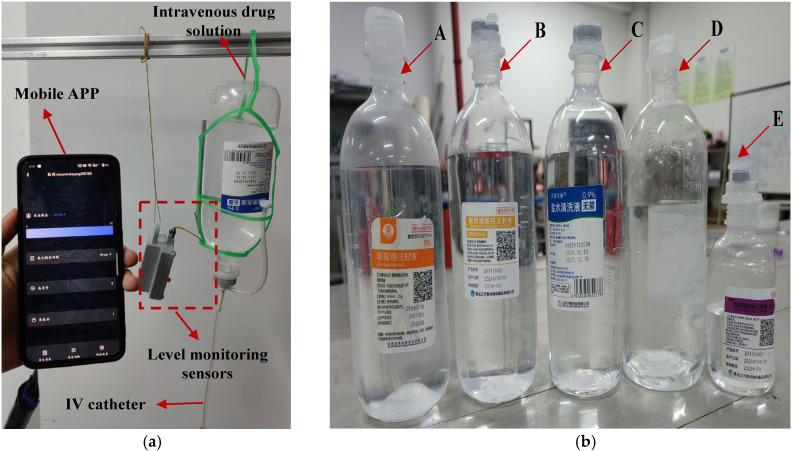
The experimental test scenario and the intravenous drug solutions used in the experiment. In (**a**), shows the test scenario for this sensor. In (**b**), A, B, C, D, and E represent five commonly used intravenous solutions: 5% glucose injection, compound sodium chloride injection, calcium gluconate injection, 0.9% sodium chloride saline, and sodium lactate Ringer’s injection.

**Figure 8 sensors-25-04524-f008:**
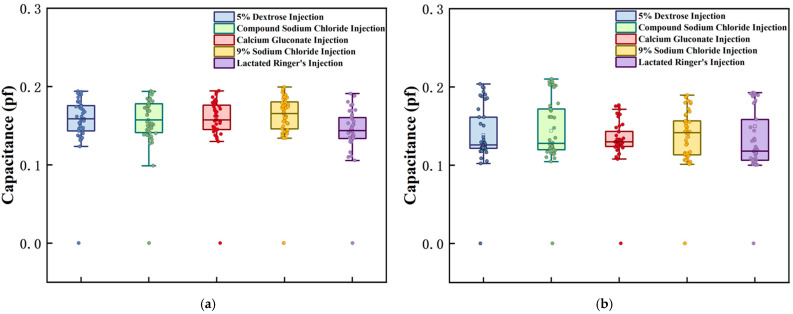
Initial capacitance test results for each intravenous drug solution. Graphs (**a**,**b**), respectively, represent the measurement results of the main pole plate and the adaptive pole plate when facing different experimental subjects.

**Figure 9 sensors-25-04524-f009:**
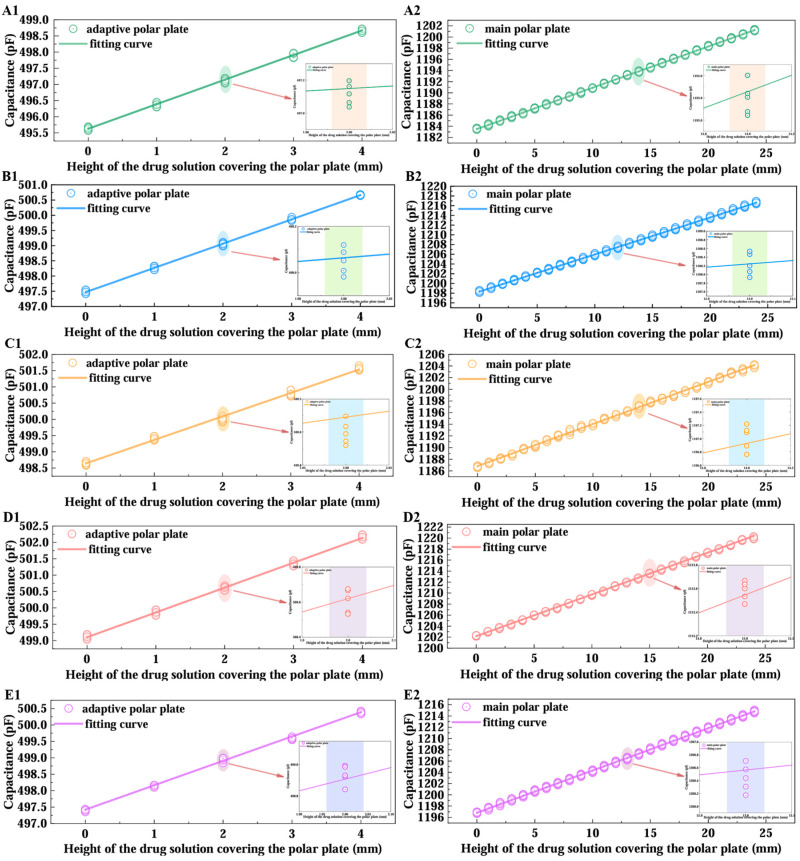
Plot of fitted curves for intravenous drug solution data. (**A**–**E**) represent sodium lactate Ringer injection, 5% dextrose injection, compound sodium chloride injection, 0.9% sodium chloride saline, and calcium gluconate injection as the experimental objects, respectively. “**1**” denotes the test results when the drug solution covers the adaptive pole plate, and “**2**” denotes the results when it covers the main pole plate.

**Figure 10 sensors-25-04524-f010:**
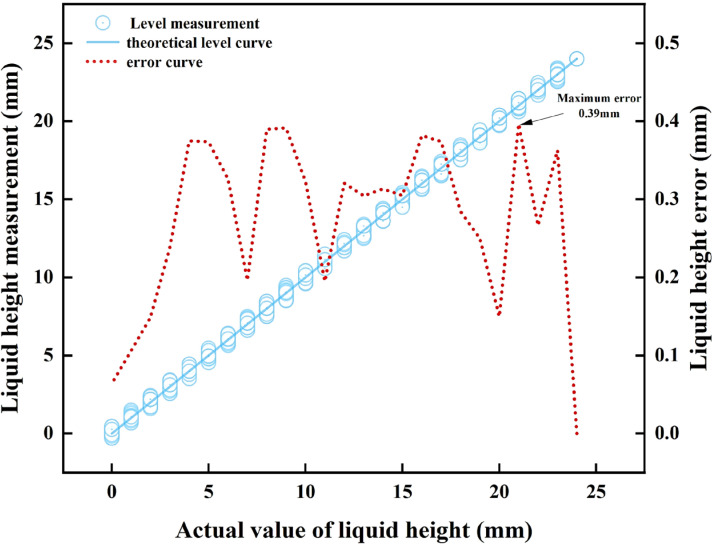
Comparison of the measured height of the drug solution with the actual height.

**Table 1 sensors-25-04524-t001:** Comparison of sensors used for IV monitoring.

Reference	[17]	[18]	[19]	[20]	This Work
Measurement Method	Capacitively coupled contactless conductivity detection (C^4^D)	Deep learning-based computer vision	Multi-sensor fusion (optical + load cell)	Optical sensing with focusing lens	Non-contact capacitive
Sensor Structure	Excitation electrode, detection electrode, shielding electrode on flexible substrate	Camera and custom neural network (for drop state classification)	Photosensor (650 nm laser + photoresistor) and load cell	LED light source, biconvex focusing lens (BFL), light-sensing module (Cds)	Flexible (FPC) double pole capacitive sensor (main pole plate, adaptive pole plate, back shielding electrode)
Resolution	0.2 mm	/	/	/	1 mm
Monitoring Accuracy Rate	100%	100%	88%	100%	100%
Calibration Required (Yes/No)	Yes	Yes	Yes	Yes	No
Cost	Very low	High	Low	High	Very low
Key Advantages	Wide range (up to 900 mm); flexible; applicable to various containers/materials	Non-invasive; full-scene recognition; suitable for telemedicine	Improved accuracy under dynamic conditions; Wi-Fi remote monitoring	Enhanced sensitivity via optical focusing; Bluetooth wireless monitoring	High sensitivity (753.5 fF/mm); calibration-free; adaptability to curved containers; excellent linearity (main pole plate: 1.99%; adaptive pole plate: 0.35%)
Key Limitations	Requires manual calibration; low sensitivity for low-dielectric materials	High computational power; sensitive to lighting conditions	Increased power consumption and integration complexity	Miniaturization challenges; risk of lens contamination	No obvious drawbacks at present

**Table 2 sensors-25-04524-t002:** Minimum measured values of initial capacitance for each drug solution.

Measurement Objects	Main Pole Plate (pF)	Adaptive Pole Plate (pF)
5% Dextrose Injection	1198.095	497.417
Compound Sodium Chloride Injection	1186.533	498.586
Calcium Gluconate Injection	1196.665	497.351
9% Sodium Chloride Injection	1202.134	499.076
Lactated Ringer’s Injection	1183.385	495.570

**Table 3 sensors-25-04524-t003:** Summary of calculated capacitive sensor parameters for diverse injectable solutions.

Intravenous Injection Solution	$k_{z}$ (pF/mm)	$k_{s}$ (pF/mm)	$\gamma_{z}$ (%)	$\gamma_{s}$ (%)	α (%)
5% Dextrose Injection	0.756	0.761	1.91%	0.41%	0.66%
Compound Sodium Chloride Injection	0.721	0.723	2.21%	0.97%	0.27%
Calcium Gluconate Injection	0.751	0.748	2.14%	0.62%	0.4%
9% Sodium Chloride Injection	0.756	0.758	1.93%	0.55%	0.27%
Lactated Ringer’s Injection	0.753	0.757	1.91%	0.42%	0.53%

## Data Availability

The data presented in this study are available on request from the first author.

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
