# Peer review of "A Novel Adaptive Flexible Capacitive Sensor for Accurate Intravenous Fluid Monitoring in Clinical Settings"

_sensors, 2025, doi:10.3390/s25144524_

Round 1
Reviewer 1 Report
Comments and Suggestions for Authors
RESEARCH MANUSCRIPT REVIEW Date
June, 2025 Journal: Sensors 2025
Title Article Adaptive Flexible IV Monitoring System
Overall. The research is interesting. I detected some very clear formatting errors that should be corrected, there is a formatting error on line 95. Those reviewing this document on behalf of the publisher should take these errors into account. As it is clear that there is a formatting error. There is another formatting error on line 147.
in Figure 268, there is a formula; I think it should be in equation format and not within the text. In Table 1, the text and numbers are formatted differently, and the font sizes are different. Table 2 contains errors; some characters are bold, and the magnitudes of these elements are not identified in the text. in general, the figures and tables should be correctly referenced. In some, the source is missing; it is not indicated whether the figures were created by the publisher or belong to a reference source. This is not indicated in some figures. All acronyms in Figure 4 should be explained in detail, for example in a table or similar. There is no chapter on the results and discussion of the research. The conclusions are very sparse; the importance and need for the research are not clarified. The funding is unclear; the country is not indicated. Important bibliography is missing, and it has not been detailed or referenced in the manuscript.
Additional comment
Some review points to consider: This article designs a flexible coplanar bipolar plate capacitance (FPC) sensor based on the parallel plate capacitance sensing mechanism and establishes the mapping relationship between the residual amount of liquid in the infusion bottle and the capacitance analog by implementing two metal-sensing polar plates of different sizes on the front of the flexible material and integrated metal protection electrodes on the back.
It is an original topic, as it contributes new features to existing research in this field.
It provides new insights into research techniques in this field. For example, the system's high-precision capacitance detection circuit, coupled with an adaptive algorithm, converts the analog capacitance into residual fluid height data, effectively eliminating the influence of disturbing factors such as the physical properties of infusion bottles and differences in the dielectric properties of solutions. It significantly improves the universality of level detection for intravenous fluids across multiple scenarios and dosage forms. Finally, five types of clinically used intravenous fluids were selected for multi-group parallel control experiments.
Improvements would include conducting more experimental tests with larger sample sizes. Are the conclusions consistent with the evidence and arguments presented and do they address the main question posed? Please also explain why this is/is not the case. The conclusions are insufficient; they should explain the results in more detail and compare them with other research.
Reviewer 2 Report
Comments and Suggestions for Authors
The manuscript presents a flexible.
Adaptive capacitive sensing system for real-time IV fluid monitoring, supported by theoretical derivation and experimental validation. The topic is relevant and has a potential impact on medical monitoring systems. However, the manuscript requires major revisions due to overly complex theoretical presentation, lack of clarity in experimental methodology. Language issues, and insufficient benchmarking. The article can be considered after addressing the following detailed concerns.
1 The authors are required to improve the connection between the theoretical model and its real world application.
Specifically, to explain how the adaptive electrode corrects for bottle material/dielectric variability in practical terms. not just via equations.
2 Please add an interpretation of the results beyond numerical fitting. What does a 1.99% linearity error mean for infusion safety?
Is the alarm delay clinically acceptable? These practical insights are currently missing.
3 Include a performance comparison (e.g., a bar chart, and table) of your system versus other common IV monitoring methods, highlighting differences in resolution, cost, calibration needs, and clinical usability.
4 The theoretical modeling section should be significantly condensed. Avoid over-deriving standard capacitor equations.
5 The authors should ensure consistent and correct SI unit formatting throughout the manuscript. Specifically, insert a non-breaking space between numerical values and units (e.g., “753.5 fF” instead of “753.5fF”). and use proper unit casing: “pF” instead of “pf”. The usage of “figure” with an initial small letter in the main text is not appropriate, as the figure descriptions start with all capital letters. Table 2 column headers should be clarified. The description of Table 2 in the main text is overly dense and lacks clarity. The authors should clearly define variables (α, γz, γs and so on). Conclude with a statement about what the table implies about sensor performance. See if the multiplication symbol in applications is proper. The text in the inset Figures of Figure 9 is not visible (also, the capacitance unit is not written properly).
- Line 95 and 97. There is Error! Reference source not found. see if this is a reference problem? The manuscript requires substantial improvement in language, grammar, and formatting to meet the standards of a peer-reviewed scientific journal.

Reviewer 3 Report
Comments and Suggestions for Authors
The reviewer comments are enclosed in the attached pdf file.

Reviewer 4 Report
Comments and Suggestions for Authors
Review report of Sensors-3719835-peer-review-v1
The manuscript describes the development and validation of a novel adaptive flexible capacitive sensing system for non-contact monitoring system of intravenous (IV) infusion levels. The physical model, sensor design, and experiments were explored. In particular, the fundamental concept of capacitance sensors was employed to formulate an effective capacitance under ideal conditions. The practical sensor and detection system were designed and fabricated. The LC resonance circuit and parasitic capacitance are introduced to take account the capacitance expression. The experimental measurement of intravenous drug solutions shows good agreement with the theoretical study. The manuscript is organized and contains interesting details. However, the proposed idea and some results presented are unclearly described. I would suggest revising the manuscript according to the below comments.
- On page 1, the title of the article is quite short and general. I would suggest the authors consider more focus on clarity and compelling. For example, “Design and Validation of an Adaptive Flexible Capacitive Sensor for Precise Intravenous Fluid Level Detection”, “A Novel Adaptive Flexible Capacitive Sensor for Accurate Intravenous Fluid Monitoring in Clinical Settings”, etc.
- On page 1, line 10, it should be “a flexible double pole capacitive (FPB) sensor” instead of “a flexible (FPB) …”
- On page 2, line 91, the Basic Theory seems focused on purely theoretical article. I would suggest “Theoretical principles”, but optional.
- On page 3, line 95, a reference is missing.
- On page 4, line 124, using an “A” for the effective area would be more appropriate than “S”.
- On page 4, line 129 and 134, typically “d” denotes the separation distance between the two plates; however, “d” with a subscript denotes the dielectric constant. I would suggest the authors consider the symbol and the expression of equation (4).
- On page 5, line 160, I would suggest this section would be “Sensor Design and Fabrication”, but optional.
- On page 7, lines 235-236, a reference (s) should be provided.
- On page 9, line 281, I would suggest using “” for sensor sensitive because the small “k” is employed in equation (13). This will help the readers more easily by following the manuscript.

Round 2
Reviewer 2 Report
Comments and Suggestions for Authors
The manuscript is accepted in the current form.